# Acute SARS-CoV-2 infections harbor limited within-host diversity and transmit via tight transmission bottlenecks

Katarina M. Braun[1], Gage K. Moreno[2], Cassia Wagner[3], Molly A. Accola[4], William M. Rehrauer[4], David A. Baker[2,5], Katia Koelle[6], David H. O'Connor[2,5], Trevor Bedford[3], Thomas C. Friedrich[1]*, Louise H. Moncla[3]*

1 Department of Pathobiological Sciences, University of Wisconsin-Madison, Madison, Wisconsin, United States of America, 2 Department of Pathology and Laboratory Medicine, University of Wisconsin-Madison, Madison, Wisconsin, United States of America, 3 Vaccine and Infectious Disease Division, Fred Hutchinson Cancer Research Center, Seattle, Washington, United States of America, 4 University of Wisconsin School of Medicine and Public Health and the William S. Middleton Memorial Veterans Hospital, Madison, Wisconsin, United States of America, 5 Wisconsin National Primate Research Center, University of Wisconsin-Madison, Madison, Wisconsin, United States of America, 6 Department of Biology, Emory University, Atlanta, Georgia, United States of America

☯ These authors contributed equally to this work.
* tcfriedrich@gmail.com (TCF); lhmoncla@gmail.com (LHM)

**Data Availability Statement:** Consensus genomes have been deposited in GISAID (gisaid.org) with accession numbers available in S1 Table. Raw Illumina reads are available in the Short Read

## Abstract

The emergence of divergent SARS-CoV-2 lineages has raised concern that novel variants eliciting immune escape or the ability to displace circulating lineages could emerge within individual hosts. Though growing evidence suggests that novel variants arise during prolonged infections, most infections are acute. Understanding how efficiently variants emerge and transmit among acutely-infected hosts is therefore critical for predicting the pace of long-term SARS-CoV-2 evolution. To characterize how within-host diversity is generated and propagated, we combine extensive laboratory and bioinformatic controls with metrics of within- and between-host diversity to 133 SARS-CoV-2 genomes from acutely-infected individuals. We find that within-host diversity is low and transmission bottlenecks are narrow, with very few viruses founding most infections. Within-host variants are rarely transmitted, even among individuals within the same household, and are rarely detected along phylogenetically linked infections in the broader community. These findings suggest that most variation generated within-host is lost during transmission.

## Author summary

RNA viruses generate diversity within individual, infected hosts. This genetic diversity can be used to trace how viruses evolve during the course of infection within individuals, and transmission between them. To investigate how SARS-CoV-2 diversity is generated and propagated, we deep sequenced 133 SARS-CoV-2 genomes isolated from acutely infected individuals in Wisconsin. We capitalize on a large dataset of consensus genomes from Wisconsin to investigate how variants are transmitted within the surrounding

Archive under bioproject PRJNA718341, accessible at https://www.ncbi.nlm.nih.gov/sra/?term=PRJNA718341. All raw Nanopore reads are available in the Short Read Archive under bioproject PRJNA614504 (accessible at https://www.ncbi.nlm.nih.gov/sra/?term=PRJNA614504). All code used to analyze the data and generate the figures shown in this manuscript are available at https://github.com/lmoncla/ncov-WI-within-host.

**Funding:** LHM is supported by NIAID grant number K99 AI147029-01. GKM is supported by an NLM training grant to the Computation and Informatics in Biology and Medicine Training Program (NLM 5T15LM007359). The funders had no role in study design, data collection and analysis, decision to publish, or preparation of the manuscript.

**Competing interests:** The authors have declared that no competing interests exist.

community, and use a unique household dataset to estimate the number of viruses that are transmitted between epidemiologically linked individuals. We find that most SARS-CoV-2 infections are characterized by limited within-host diversity, and that the vast majority of intra-host single nucleotide variants (iSNVs) are lost during transmission. We do not find evidence that variation is frequently propagated along phylogenetically linked infections, and estimate that most infections are founded by very few unique virions. The combination of limited within-host diversity and tight transmission bottlenecks may slow the pace of SARS-CoV-2 evolution in the future, and suggests that extensive within-host evolution is likely rare.

## Introduction

The recent emergence of variants of concern has spurred uncertainty about how severe acute respiratory coronavirus 2 (SARS-CoV-2) will evolve in the longer term. SARS-CoV-2 acquires a fixed consensus mutation approximately every 11 days as it replicates in a population [1]. However, lineages of SARS-CoV-2 have arisen harboring more variants than expected based on this clock rate, with some variants rapidly displacing existing circulating lineages and/or conferring antibody escape [2,3]. The emergence of these lineages has raised concern that SARS-CoV-2 may rapidly evolve to evade vaccine-induced immunity, and that vaccines may need to be frequently updated. A current leading hypothesis posits that these lineages may have emerged during prolonged infections. Under this hypothesis, longer infection times, coupled with antibody selection [4], may allow more time for novel mutations to be generated and selected before transmission. Studies of SARS-CoV-2 [4–8] and other viruses [9,10] support this hypothesis. Longitudinal sequencing of SARS-CoV-2 from immunocompromised or persistently infected individuals accordingly reveals an accumulation of intrahost single-nucleotide variants (iSNVs) and short insertions and deletions (indels) during infection [4–6,11]. In influenza virus and norovirus infections, variants that arose in immunocompromised patients were later detected globally, suggesting that long-term infections may mirror global evolutionary dynamics [9,12]. Mutations defining novel variant lineages resulting in lineage displacement and/or immune escape in SARS-CoV-2 Spike, like Δ69/70, N501Y and E484K, have already been documented arising in persistently infected and immunocompromised individuals [4,5].

While prolonged infections occur, the vast majority of SARS-CoV-2 infections are acute [13]. Viral evolutionary capacity is limited by the duration of infection [14], and it is not yet clear whether the evolutionary patterns observed during prolonged SARS-CoV-2 infections also occur commonly in acutely infected individuals. Replication-competent virus has rarely been recovered from individuals with mild to moderate coronavirus disease 2019 (COVID-19) beyond ~10 days following symptom onset [15,16]. Multiple studies of influenza viruses show that immune escape variants are rarely detected during acute infection, even within vaccinated individuals [17–19]. Detailed modeling of influenza dynamics suggests that the likelihood of within-host mutation emergence depends on the interplay of immune response timing, the de-novo mutation rate, and the number of virus particles transmitted between hosts [14]. Understanding the speed with which SARS-CoV-2 viruses acquire novel mutations that may escape population immunity will be critical for formulating future vaccine updates. If novel immune-escape variants emerge primarily within long-term infections, then managing long-term infections in an effort to reduce any onward transmission may be critical. Conversely, if

novel variants are efficiently selected and transmitted during acute infections, then vaccine updates may need to occur frequently.

A clear consensus on how frequently variants are shared and transmitted between individuals has been elusive. Estimates of SARS-CoV-2 diversity within hosts have been highly variable, and comparing results among labs has been complicated by sensitivity to variant-calling thresholds and inconsistent laboratory controls [20–23]. Data suggests that SARS-CoV-2 genetic diversity within individual hosts during acute infections is limited [20,24] and shaped by genetic drift and purifying selection [21,25–27]. Estimates of the size of SARS-CoV-2 transmission bottlenecks [21,28,29] have ranged considerably, and recent validation work has shown that estimates of within-host diversity and transmission bottleneck sizes are highly sensitive to sequencing protocols and data analysis parameters, like the frequency cutoff used to identify true within-host variants [20,30]. Clarifying the extent to which within-host variants arise and transmit among acutely infected individuals, while controlling for potential error, is critical for assessing the speed at which SARS-CoV-2 evolves and adapts.

To characterize how within-host variants are generated and propagated, we employ extensive laboratory and bioinformatic controls to characterize 133 SARS-CoV-2 samples collected from acutely-infected individuals in Wisconsin, United States. Unlike other existing studies, we explicitly designed our investigation with controlling for bioinformatic and laboratory error in mind, sequencing every sample in technical replicate and validating our variant calls with an entirely separate bioinformatic pipeline. By comparing patterns of intrahost single nucleotide variants (iSNVs) to densely-sampled consensus genomes from the same geographic area, we paint a clear picture of how variants emerge and transmit within communities and households. We find that overall within-host diversity is low during acute infection, and that iSNVs detected within hosts almost never become dominant in later-sampled sequences. We find that iSNVs are infrequently transmitted, even between members of the same household, and we estimate that transmission bottlenecks between putative household pairs are narrow. This suggests that most iSNVs are transient and very rarely transmit beyond the individual in which they have originated. Our results imply that during typical, acute SARS-CoV-2 infections, the combination of limited intrahost genetic diversity and narrow transmission bottlenecks may slow the pace by which novel variants arise, are selected, and transmit onward. Finally, our findings are consistent with the hypothesis that novel variants are more likely to be selected to high frequencies during the course of prolonged infections, and that a minority of infections, either acute or prolonged, may play outsize roles in the emergence of novel variants. Followup studies that continue to dissect the degree with which individual infections, including both acute and chronic, contribute to global evolution will be necessary for confirming this hypothesis. Finally, future examination of how SARS-CoV-2 evolution proceeds in individuals with prior exposure via vaccination or infection will be necessary for extending these results as SARS-CoV-2 continues to circulate and evolve.

## Results

### Within-host variation is limited and sensitive to iSNV-calling parameters

Viral sequence data provide rich information about how variants emerge within, and transmit beyond, individual hosts. Viral nucleotide variation generated during infection provides the raw material upon which selection can act. However, viral sequence data are sensitive to multiple sources of error [20,22,23], which has obscured easy comparison among existing studies of SARS-CoV-2 within-host evolution. Here, we take several steps to minimize sources of error and to assess the robustness of our results against variable within-host single nucleotide variant (iSNV)-calling parameters.

We identified spurious iSNVs introduced by our library preparation pipeline by sequencing in duplicate a clonal, synthetic RNA transcript identical to our reference genome (MN90847.3). We considered only variants found in both technical replicates, which we refer to as "intersection iSNVs". We detected 7 intersection iSNVs at $\geq$1% frequency (**S1 Table**); 2 of these were previously identified by a similar experiment in Valesano et al. [20]. We excluded all 7 of these iSNVs from downstream analyses. To exclude laboratory contamination, we sequenced a no-template control (water) with each large sequencing batch and confirmed that these negative controls contained <10x coverage across the SARS-CoV-2 genome (**S1 and S2 Figs**). To ensure that spurious variants were not introduced by our bioinformatic pipelines, we validated our iSNV calls using a second pipeline which employs distinct trimming, mapping, and variant calling softwares. We found near-equivalence between the two pipelines' iSNV calls ($R^2$ = 0.998; **S3A Fig**), providing additional independent support for our bioinformatic pipeline to accurately call iSNVs.

Viral iSNV calls are also sensitive to the variant-calling threshold (i.e., a minimum frequency at which iSNVs must occur to be considered non-artefactual) applied [22] and the number of viral input copies. Work by Grubaugh et al. [31] showed highly accurate iSNV calls with tiled amplicon sequencing using technical replicates and a 3% frequency threshold. Consistent with this observation, we observed a near-linear correlation between iSNVs called in each replicate at a 3% frequency threshold ($R^2$ = 0.992) (**Fig 1A**). Unsurprisingly, we find the proportion of intersection iSNVs compared to all iSNVs within a given sample increases as the frequency threshold increases (**S3B Fig**). Additionally, 57/102 iSNVs detected in our clonal RNA controls occur <3% frequency in a single replicate (**S3C Fig**). We detected no intersection iSNVs <3% frequency.

Consistent with previous studies, we observed a negative correlation between Ct and the overlap in variants between replicates such that high-Ct (i.e., low vRNA copy number) samples had fewer intersection iSNVs called in each replicate (**Fig 1B**) [22,31]. Although we do not have access to absolute quantification for viral input copies for our sample set, we can use results of semi-quantitative clinical assays on the sequenced specimens as a proxy for viral RNA (vRNA) concentration. Samples with low viral RNA copy numbers can sometimes result in an excess of spuriously detected iSNVs [22], so we wanted to ensure that we did not detect a greater number of iSNVs in samples in our dataset with low RNA concentrations. Using input data from two different clinical assay platforms, we find no correlation between viral input copies and the number of intersection iSNVs detected, consistent with findings reported by others, and supporting the robustness of our iSNV calls to RNA copy numbers [21,32] (**S3D and S3E Fig**).

Based on these observations, we chose to use a 3% iSNV frequency cutoff for all downstream analyses, and report only iSNVs that were detected in both technical replicates, at a frequency $\geq$3%. Using these criteria, we found limited SARS-CoV-2 genetic diversity in most infected individuals: 22 out of 133 samples did not harbor even a single intersection iSNV at $\geq$3% frequency. Among the 111 samples that did harbor within-host variation, the average number of iSNVs per sample was 3.5 (standard deviation = 2.6, median = 3, range = 1–11) (**Fig 1C**). Most iSNVs were detected at <10% frequency (**Fig 1D**). Compared to expectations under a neutral model, every type of mutation we evaluated (synonymous, nonsynonymous, intergenic region, and stop) was present in excess at low frequencies, consistent with purifying selection or population expansion within the host (**Fig 1D**). Taken together, our results confirm that the number of iSNVs detected within-host are dependent on variant-calling criteria. Once rigorous laboratory and bioinformatic controls are applied, we find that most infections during the spring 2020 are characterized by 0–6 iSNVs, primarily detectable at $\leq$10% frequency.

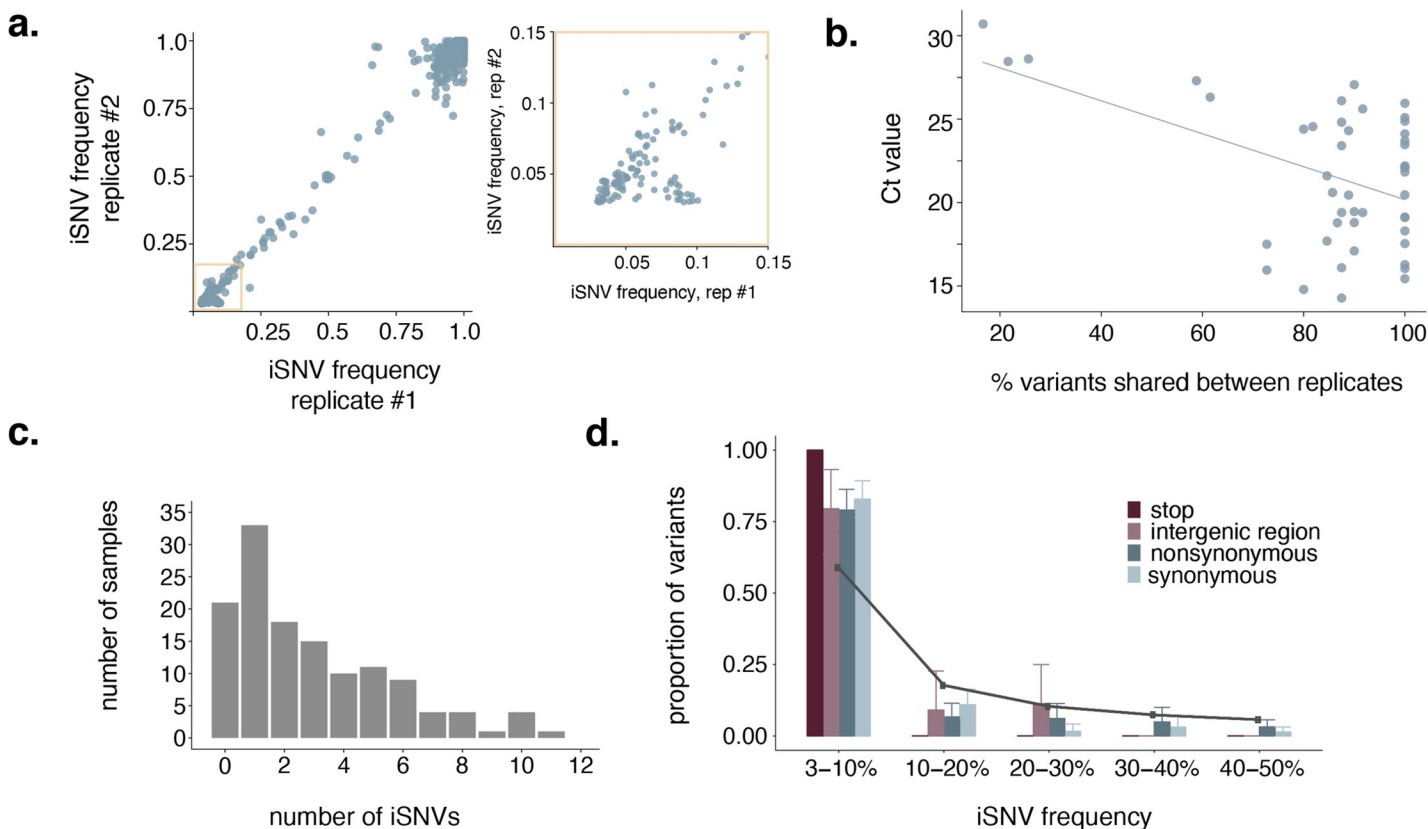

**Fig 1. Within host variation is limited after data quality control. a.** iSNV frequencies in replicate 1 are shown on the x-axis and frequencies in replicate 2 are shown on y-axis. The yellow box highlights low-frequency iSNVs (3–15%), which is expanded out to the right. **b.** The Ct value is compared to the percent of iSNVs shared between technical replicates. The blue line is a line of best fit to highlight the observed negative trend. **c.** Distribution of the number of total iSNVs detected per sample. 22 out of 133 samples harbor no iSNVs at all, and the maximum number of iSNVs in a single sample was 11. **d.** The proportion of iSNVs that were detected at various within-host frequency bins is shown. Error bars represent the variance in the proportion of total within-host iSNVs within that frequency bin across samples in the dataset as calculated by bootstrapping. There was a single stop variant in the entire dataset, so no error bar is shown for the stop category. The solid grey line indicates the expected proportion of variants in each frequency bin under a neutral model.

## Recurrent iSNVs consist of Wuhan-1 reversions and common polymorphic sites

Previous studies of SARS-CoV-2 evolution have noted the unusual observation that iSNVs are sometimes shared across multiple samples. Understanding the source and frequency of shared iSNVs is important for measuring the size of transmission bottlenecks and for identifying potential sites of selection. In our dataset, 143/184 (77%) iSNVs were unique to a single sample (**Fig 2A**). However, 41 iSNVs were detected in at least 2 samples. These "shared iSNVs" were detected across multiple sequencing runs (**S5 Fig**), and were absent in our negative controls, suggesting they are unlikely to be artefacts of method error. Most of the shared iSNVs we detect fall into two categories: iSNVs that occur within or adjacent to a homopolymer region (8/41 iSNVs, **Fig 2B**, yellow and purple bars), or iSNVs that represent "Wuhan-1 reversions" (31/41 iSNVs, **Fig 2B**, blue and purple bars). We classified iSNVs as "Wuhan-1 reversions" when a sample's consensus sequence had a near-fixed variant (50–97% frequency) relative to the Wuhan-1 reference, with the original Wuhan-1 nucleotide present as an iSNV. For example, if a consensus change from C to T was found at 95%, we called the C at 5% an iSNV in our dataset. iSNVs in or near homopolymer regions were defined as those that fall within or one nucleotide outside of a span of at least 3 identical nucleotide bases. Shared iSNVs were detected

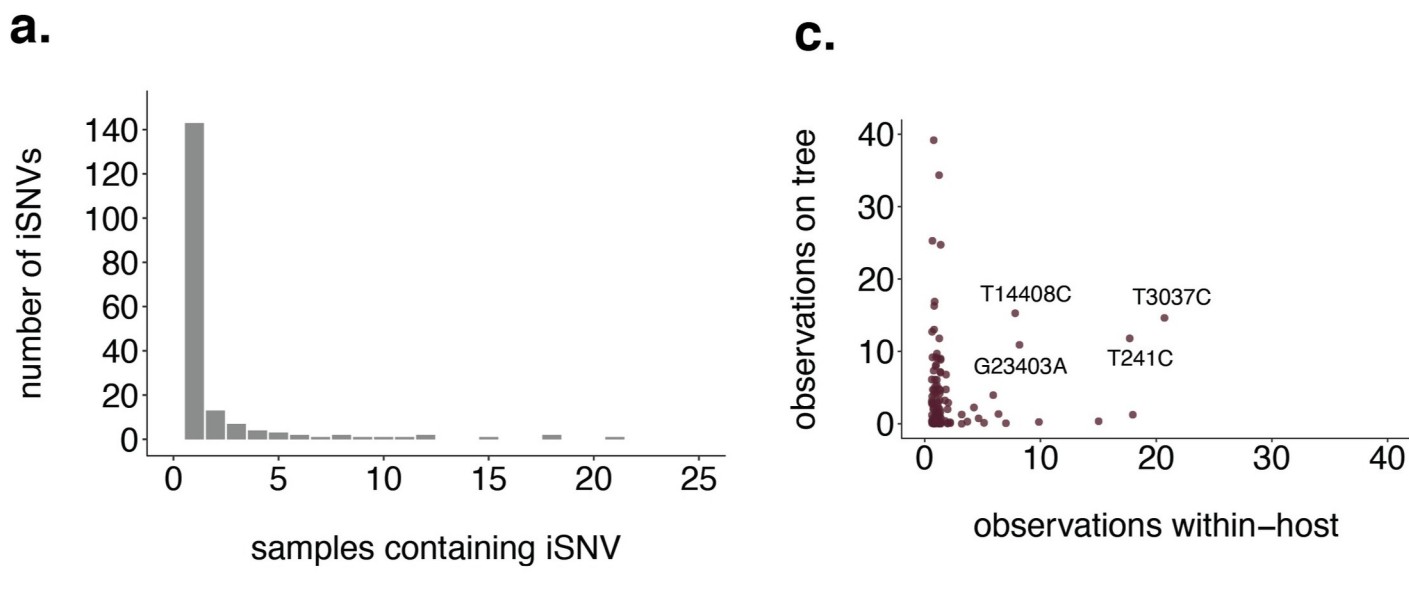

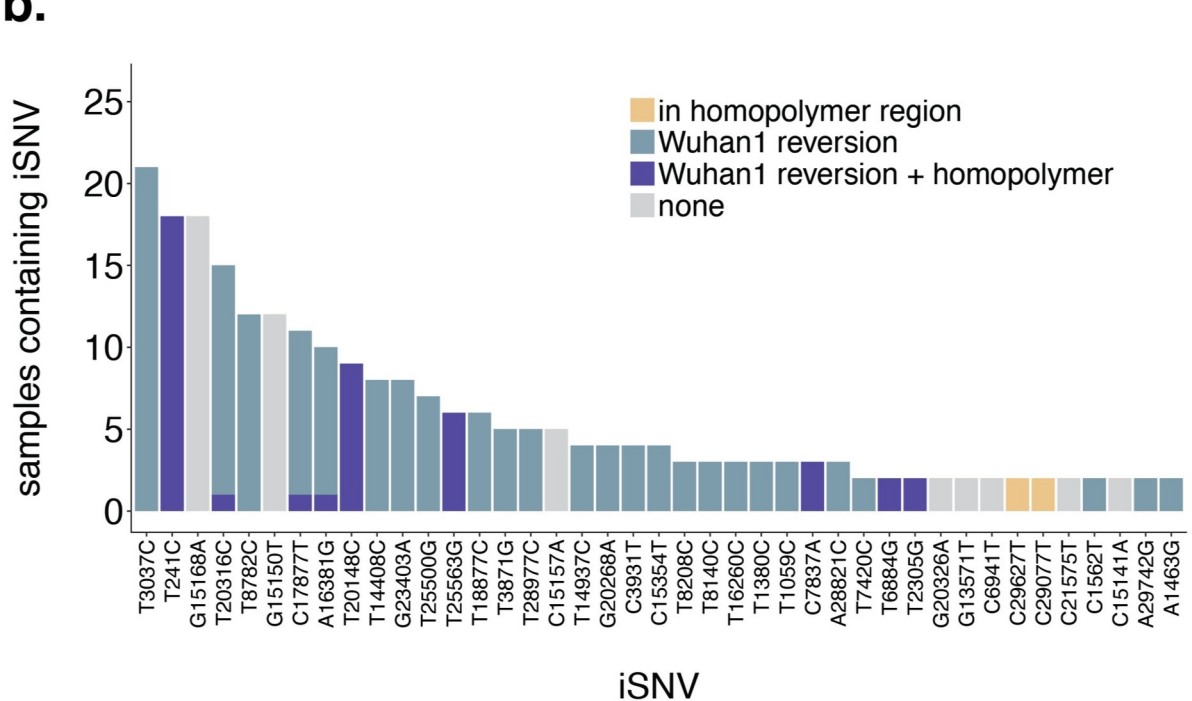

**Fig 2. Shared iSNVs represent homopolymers and common polymorphic sites. a.** The number of iSNVs (y-axis) present within n individuals (x-axis) is shown. 143/184 (77%) of iSNVs are found in only a single sample. 6 iSNVs are shared by at least 10 samples. **b.** Each iSNV detected in at least 2 samples is shown. Variants that occur within, or 1 nucleotide outside of, a homopolymer region (classified as a span of the same base that is at least 3 nucleotides long) are colored in yellow. Variants that represent the minor allele for variants that were nearly fixed at consensus (annotated here as "Wuhan1 reversions") are shown in blue, and variants that were both Wuhan1 reversions and occurred in homopolymer regions are colored in purple. **c.** For each unique iSNV detected within a host, the x-axis represents the number of samples in which that iSNV was detected, and the y-axis represents the number of times it is present on the global SARS-CoV-2 phylogenetic tree. The counts on the phylogenetic tree represent the number of times the mutation arose along internal and external branches. The variants labeled with text are those that are detected at least 5 times within-host and at least 5 times on the phylogeny. Two of the most commonly detected iSNVs, T3037C and T241C (shown as the furthest to the left in panel b), are also frequently detected on the phylogenetic tree.

in homopolymer regions a total of 44 times across samples, with strong enrichment in A/T homopolymers (39/45 detections) compared to G/C homopolymers (6/45 detections). **S5 Table** lists all iSNV frequencies, genome locations, and protein changes (amino acid changes) detected in our dataset. A visualization of this information is shown in **S12 Fig.** Overall, this suggests that shared variants in our dataset may be at least partially explained by viral polymerase incorporation errors, potentially in A/T-rich regions, and at sites that are frequently polymorphic.

The most commonly detected iSNVs in our dataset represent Wuhan-1 reversions at nucleotide sites 241 (detected 18 times; within/adjacent to a homopolymer region) and 3037 (detected 21 times; not in a homopolymer region). Both of these sites are polymorphic deep in the SARS-CoV-2 phylogeny near the branch point for clade 20A (Nextstrain clade nomenclature). Within-host polymorphisms at sites 241 and 3037 were also detected in recent studies in the United Kingdom and Austria [21,28]. T241C and T3037C are both synonymous variants, and have emerged frequently on the global SARS-CoV-2 phylogenetic tree, suggesting that these sites may be frequently polymorphic within and between hosts across multiple geographic areas (**Fig 2C**).

## Within-host variants are found only once in phylogenetically linked infections

The emergence of divergent SARS-CoV-2 lineages has raised concerns that new variants may be selected during infection and efficiently transmitted onward. We next sought to characterize whether iSNVs arising within hosts contribute to consensus diversity sampled later in time. Using the Wisconsin-specific phylogenetic tree (**S6 Fig**), we queried whether iSNVs detected within hosts are ever found at consensus in tips sampled downstream. For each Wisconsin tip that lay on an internal node and for which we had within-host data, we traversed the tree from that tip to each subtending tip. We then enumerated each mutation that occurred along that path, and compared whether any mutations that arose on downstream branches matched iSNVs detected within-host (see **Fig 3A** for a schematic). Of the 110 Wisconsin tips harboring within-host variation, 93 occurred on internal nodes. Of those, we detect only a single instance in which an iSNV detected within a host was later detected at consensus. C1912T (a synonymous variant) was present in USA/WI-UW-214/2020 at ~4% frequency, and arose on the branch leading to USA/WI-WSLH-200068/2020 (**Fig 3B**). USA/WI-UW-214/2020 is part of a large polytomy, so this does not necessarily suggest that USA/WI-UW-214/2020 and USA/WI-WSLH-200068/2020 fall along the same transmission chain. These results indicate that despite relatively densely sampling of consensus genomes from related viruses from Wisconsin (1% of all confirmed cases as of February 16, 2021), we do not find evidence that iSNVs frequently rise to consensus along phylogenetically linked infections.

If iSNVs arising during infection are adaptive and efficiently transmitted, then they should be enriched on internal nodes of the phylogenetic tree. For each within-host variant detected in our dataset, we queried the number of times it occurred on the global SARS-CoV-2 phylogeny on tips and internal nodes. We then compared the ratio of detections on tips vs. internal nodes to the overall ratio of mutations on tips vs. internal nodes on the phylogeny. 42% (77/184) of iSNVs are present at least once at consensus level on the global phylogeny (**S7 Fig**). Of the iSNVs from our dataset that also occur in consensus genomes on the global tree, only 15 are found at least 10 times (**Figs 3C** and **S7**). iSNVs that are also found at consensus are present on internal nodes and tips at a ratio similar to that of consensus mutations overall (ratio of mutations on phylogeny nodes:tips = 4,637:17,200; ratio of iSNVs on nodes:tips = 128:411, $p = 0.16$, Fisher's exact test). Although this is the predominant pattern, we detect one

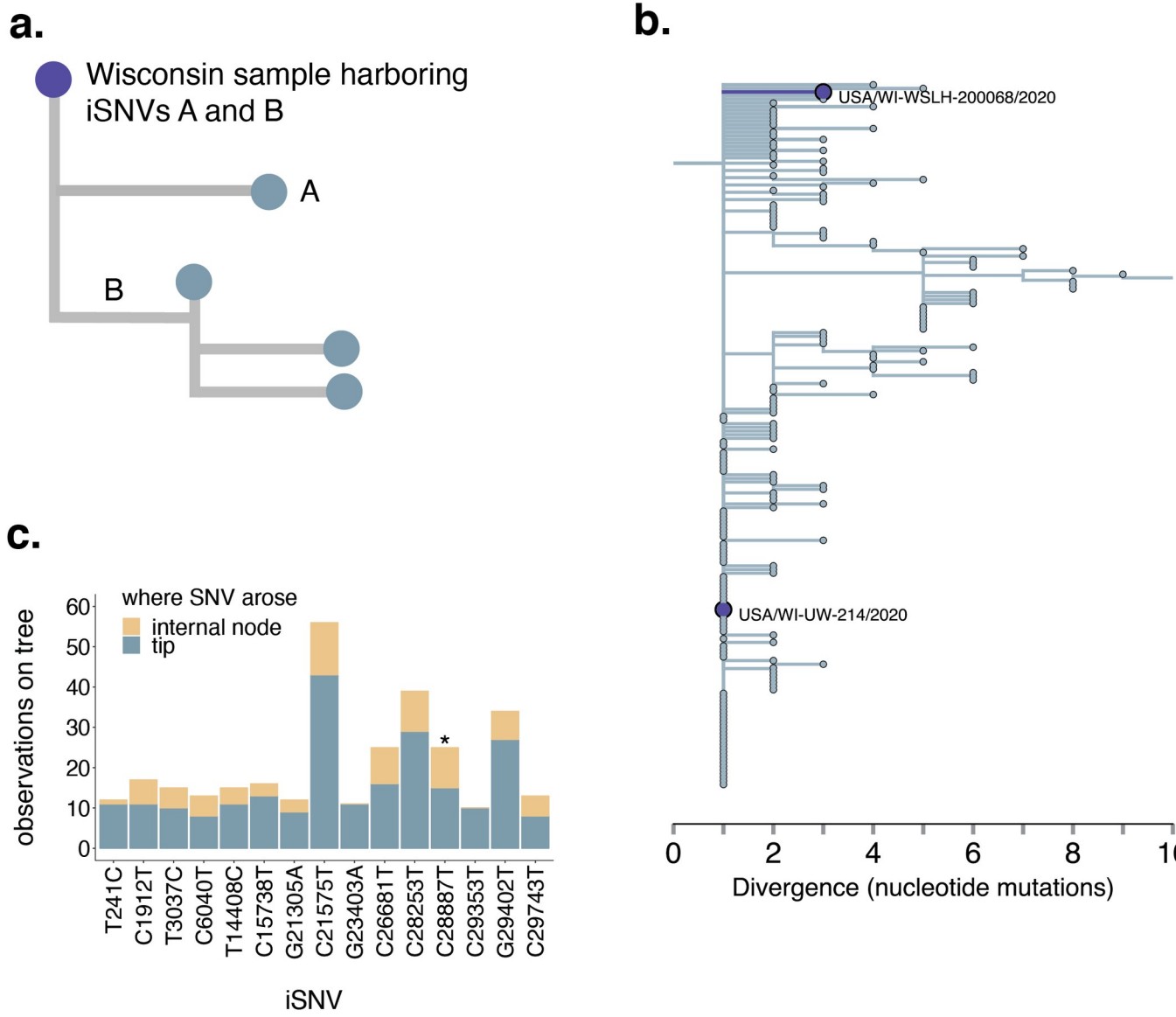

**Fig 3. Variants are not common in consensus sequences or in downstream branches. a.** We traversed the Wisconsin-focused full-genome SARS-CoV-2 phylogeny from root to tip. For each Wisconsin tip for which we had within-host data, we queried whether any of the iSNVs detected in that sample were ever detected in downstream branches at consensus. In this example, the purple tip represents a Wisconsin sample for which we have within-host data. This sample harbors 2 iSNVs, A and B. iSNV A arises on a tip that falls downstream from the starting, purple tip. iSNV B is present on a downstream branch leading to an internal node. Both A and B would be counted as instances in which an iSNV was detected at consensus in a downstream branch. **b.** In the Wisconsin-specific phylogenetic tree, we applied the metric described in **a.** Among 110 Wisconsin samples that harbored within-host variation, 93 occurred on internal nodes. Of those, we detect one instance in which a mutation detected as an iSNV in one sequence was detected in a downstream consensus sequence. (C1912T, an iSNV in USA/WI-UW-214/2020, was detected downstream in USA/WI-WSLH-200068/2020.) **c.** For each iSNV identified in the study (in at least 1 sample), we enumerated the number of times that variant occurred on the global SARS-CoV-2 phylogeny on an internal node (yellow) or on a tip (blue). The results for every variant are shown in **S6 Fig**. Here, we show only the variants that were detected at least 10 times on the global phylogeny. Each such iSNV is found at internal nodes and tips at a ratio comparable to overall mutations on the tree, except for C28887T, which is enriched on internal nodes (p = 0.028, Fishers' exact test). * indicates p-value < 0.05.

exception. C28887T is present in one sample in our dataset at a frequency of ~6%, but is found on 10 internal nodes and 15 tips (p = 0.028, Fisher's exact test) (**Fig 3C**). C28887T encodes a threonine-to-isoleucine change at position 205 in the N protein, and is a clade-defining mutation for the B.1.351 lineage. Although the functional impact of this mutation is not completely

understood, N T205I may increase stability of the N protein [33,34]. Despite the detection within-host and subsequent emergence of N205I globally, this iSNV was only detected in our dataset in one sample at low frequency. In general, across our dataset, the frequency with which iSNVs were detected within-host vs. on the phylogenetic tree is not correlated (**Fig 2C**). This suggests that although putative functional mutations may arise within a host, we do not see evidence for most iSNVs as selectively beneficial, in agreement with patterns observed in other viral pathogens [35].

## Shared iSNVs are insufficient for resolving transmission linkage

Household studies provide the opportunity to investigate transmission dynamics in a setting of known epidemiologic linkage. We analyzed 44 samples collected from 19 households from which multiple individuals were infected with SARS-CoV-2 (more information on household structure, accession numbers, and days post symptom onset can be found in **S4 Table**). Only a single timepoint from each individual, with collection times relative to their symptomatic date ranging from asymptomatic or never symptomatic to 15 days after symptom onset, were collected and analyzed for this study. To define putative transmission pairs from our household dataset, we modeled the expected number of mutations that should differ between consensus genomes given one serial interval as previously described [36,37](see Materials and Methods for details and rationale). We estimate that members of a transmission pair should generally differ by 0 to 2 consensus mutations (**Fig 4A**), and classify all such pairs within a household as putative transmission pairs. While most samples derived from a single household had near-identical consensus genomes, we observed a few instances in which consensus genomes differed substantially. In particular, USA/WI-UW-476/2020 differed from both other genomes from the same household by 11 mutations, strongly suggesting that this individual was independently infected.

To determine whether putative household transmission pairs shared more within-host variation than randomly sampled pairs of individuals, we performed a permutation test. We randomly sampled individuals with replacement and computed the proportion of iSNVs shared among random pairs to generate a null distribution (**Fig 4B**, grey bars). We then computed the proportion of variants shared among each putative household transmission pair. Finally, we compared the distribution of shared variants among household pairs and random pairs (**Fig 4B**). 90% of random pairs do not share any iSNVs. Although household pairs share more iSNVs than random pairs on average, half (14/28) of all household pairs share no iSNVs at all. Only 7 out of 28 of household pairs share more iSNVs than expected by chance (p < 0.05).

While we hypothesized that putative transmission linkage would be the best predictor of sharing iSNVs, other processes could also result in shared iSNVs. For example, if transmission bottlenecks are wide and iSNVs are efficiently transmitted along transmission chains, then iSNVs may be propagated during community transmission. If so, then iSNVs should be shared among samples that are phylogenetically close together. If transmission chains circulate within local geographic areas, then iSNVs may be commonly shared by samples from the same geographic location. Finally, if iSNVs are strongly constrained by genetic backbone, then variants may be more likely to be shared across samples from the same clade.

To measure the contribution of these factors, we computed the proportion of iSNVs shared by each pair of samples in our dataset (including household and non-household samples), and model the proportion of shared iSNVs as the combined effect of phylogenetic divergence between the tips (i.e., the branch length in mutations between tips), clade membership, geographic distance between sampling locations, and household membership. Phylogenetic divergence and geographic distance between sampling locations have minimal predicted impact on

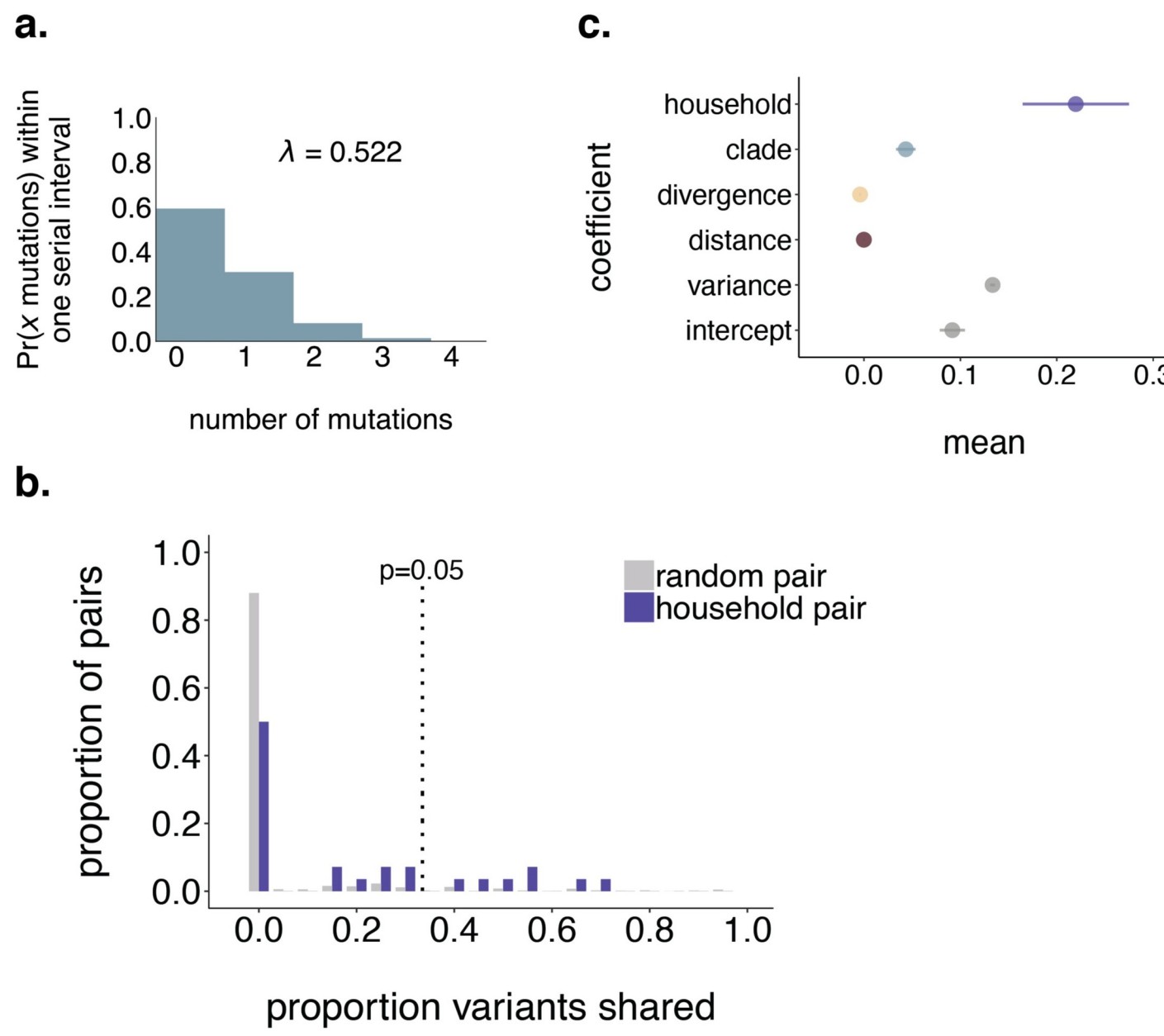

**Fig 4. A quarter of household pairs share more iSNVs than random expected by chance. a.** We modeled the probability that 2 consensus genomes will share x mutations as Poisson-distributed with lambda equal to the number of mutations expected to accumulate in the SARS-CoV-2 genome over 5.8 days [37] given a substitution rate of $1.10 \times 10^{-3}$ substitutions per site per year [1]. Exploration of how these probabilities change using a range of plausible serial intervals and substitution rates is shown in **S8 Fig.** The vast majority of genomes that are separated by one serial interval are expected to differ by $\leq 2$ consensus mutations. **b.** The proportion of random pairs (grey) and putative household transmission pairs (purple) is shown on the y-axis vs. the proportion of iSNVs shared. The dotted line indicates the 95th percentile among the random pairs. Household pairs that share a greater proportion of iSNVs than 95% of random pairs (i.e., are plotted to the right of the dotted line) are considered statistically significant at p = 0.05. iSNVs had to be present at a frequency of $\geq 3\%$ to be considered in this analysis. **c.** We assessed the impact of household membership, clade membership, phylogenetic divergence, and geographic distance on the proportion of iSNVs shared between each pair of samples in our dataset. The mean of each estimated coefficient in the combined linear regression model including all predictors is shown on the x-axis, with lines of spread indicating the range of the estimated 95% highest posterior density interval (HPDI).

iSNV sharing (**Figs 4C and S9**). The strongest predictor of sharing iSNVs is being sampled from the same household, which increased the predicted proportion of shared iSNVs by 0.22 (0.16–0.27, 95% HPDI). Belonging to the same clade increases the predicted proportion of

shared iSNVs by 0.043 (0.033–0.053, 95% HPDI), likely because sharing a within-host variant is contingent on sharing the same consensus base. Taken together, being sampled from the same household is the strongest predictor of sharing iSNVs, and 25% of household pairs share more variation than expected by chance. However, the presence of shared iSNVs alone is insufficient for inferring transmission linkage independent of additional epidemiologic data.

## Transmission bottlenecks are narrow, and sensitive to variant calling threshold

The number of viral particles that found infection is a crucial determinant of the pace at which novel, beneficial variants can emerge. Narrow transmission bottlenecks can induce a founder effect that purges low-frequency iSNVs, regardless of their fitness. Conversely, wide transmission bottlenecks result in many viral particles founding infection, reducing the chance that beneficial variants are lost. Understanding the size of the transmission bottleneck is therefore important for evaluating the probability that novel SARS-CoV-2 variants arising during acute infection will be transmitted onward. While the above permutation test compares the presence and absence of iSNVs between individuals, it does not assume transmission directionality or make use of variant frequencies. It is therefore a crude metric of whether samples share iSNVs, rather than a quantitative measure of transmission stringency. We therefore next inferred transmission bottleneck sizes using the beta-binomial inference method [38]. We inferred transmission directionality using the date of symptom onset or date of sample collection (see Materials and Methods for details). If this information was not informative, we calculated a bottleneck size bi-directionally evaluating each individual as the possible donor. In this method, the unit of the transmission bottleneck is the number of viruses that found infection in a recipient host following transmission. In total, we performed 40 transmission bottleneck size estimates in 28 putative household pairs.

iSNV frequencies in donor and recipient pairs are plotted in **Fig 5A**. Most iSNVs detected in the donor are either lost or fixed following transmission in the recipient. However, there are a few low-frequency and near-fixed iSNVs which are shared in donor-recipient pairs. The combined maximum likelihood estimate for mean transmission bottleneck size at our defined 3% frequency threshold is 15 (95% CI: 11–21), although results vary across pairs (**Fig 5B**). Prior transmission bottleneck estimates have changed based on the variant-calling threshold employed [28,30]. To determine whether our estimates were sensitive to our choice of a 3% variant threshold, we evaluated bottleneck sizes using variant thresholds ranging from 1% to 20%. We estimate the highest mean transmission bottleneck size when we employ a 1% frequency threshold (38, 95% CI: 33–43), and lowest when we use a $\geq$7% frequency threshold (2, 95% CI: 1–4) (**Figs 5C and S10**). The finding of larger bottleneck sizes at a 1% threshold may be due to increased false-positive iSNVs at lower thresholds, in agreement with our findings that a majority (56%) 57/102 iSNVs detected in the clonal RNA control occurred at frequencies <3% in a single replicate. Importantly though, while the variant threshold clearly impacts estimated bottleneck size, bottleneck size estimates range from 2–43 and never exceed 50 across a wide range of frequency thresholds.

The beta-binomial inference method assumes that shared variation in donor-recipient pairs is due to transmission. However, it is possible that shared low-frequency iSNVs are recurring mutations (i.e. homoplasies) that should be excluded from the beta-binomial analysis. One site in particular, a synonymous change at nucleotide 15,168 in ORF1ab, was commonly found at low frequencies in donor-recipient pairs. To account for the possibility that this variant is a homoplasy rather than shared via transmission, we dropped this site from our dataset and re-calculated bottleneck sizes. While bottleneck size estimates decrease in individual pairs where

**a.**

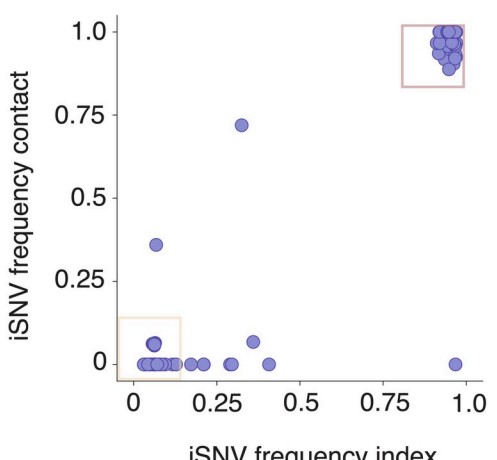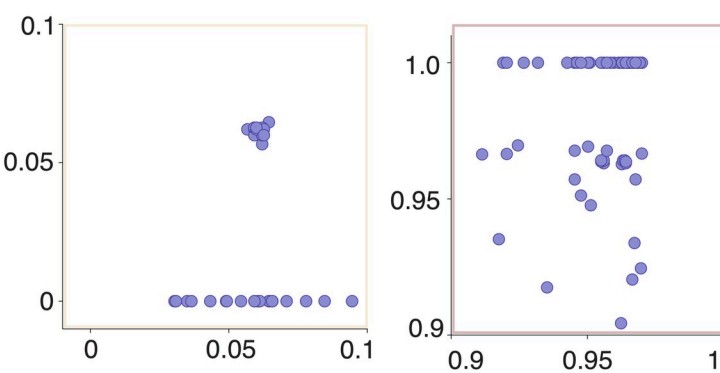

**b.**

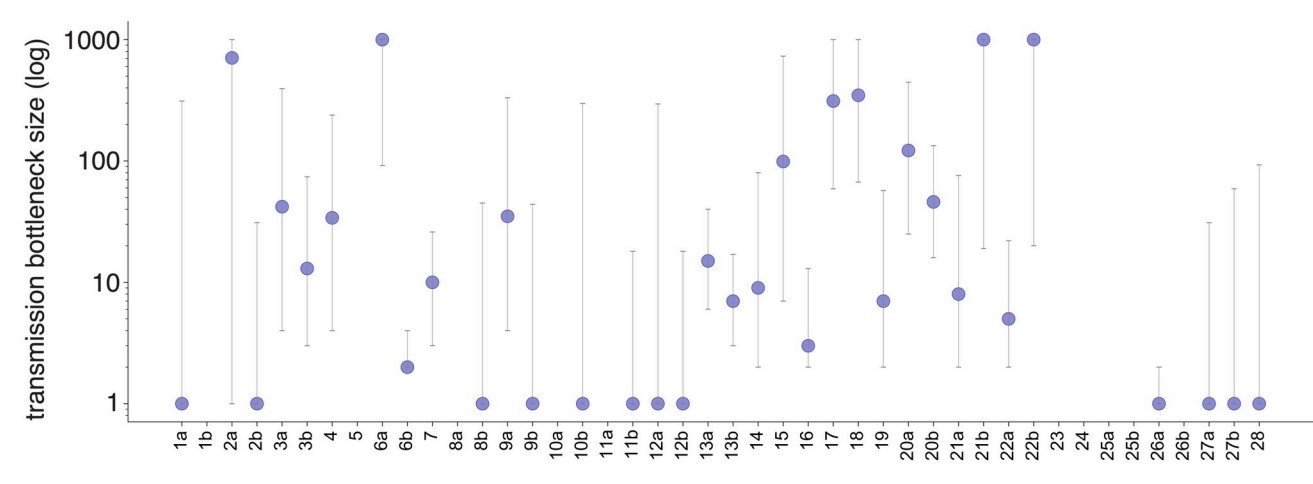

**c.**

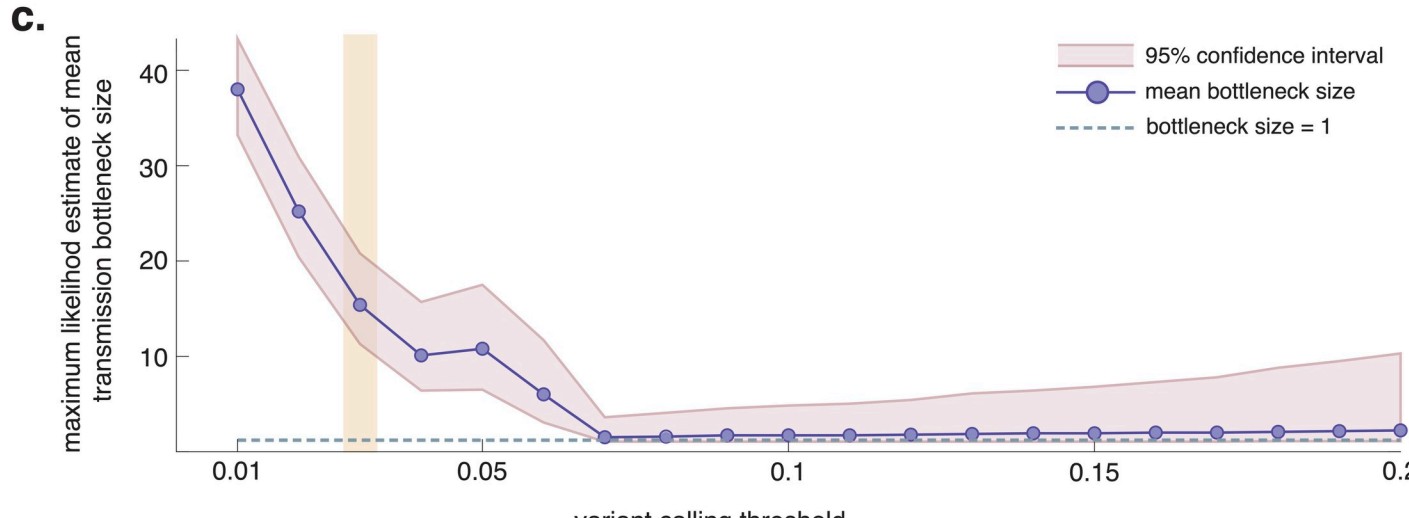

**Fig 5. SARS-CoV-2 transmission bottlenecks in household transmission pairs. a.** "TV plots" showing intersection iSNV frequencies in all 44 donor-recipient pairs using a 3% frequency threshold. The yellow box highlights low-frequency iSNVs (3–10%) and the mauve box highlights high-frequency iSNVs (90–100%). **b.** Maximum likelihood estimates for mean transmission bottleneck size in individual donor-recipient pairs. Bottleneck sizes could not be estimated for a few pairs (e.g. pairs 5, 10a, 11a, etc) because there were no polymorphic sites detected in the donor. **c.** Bidirectional comparisons are denoted with an "a" and "b" following the pair number. Combined maximum likelihood estimates across all 44 donor-recipient pairs plotted against variant calling thresholds ranging from 1–20%. The purple line shows combined estimates at each variant calling threshold shown and the mauve band displays the 95% confidence interval for this estimate. The dashed grey line indicates a bottleneck size equal to 1. The vertical yellow band highlights the combined transmission bottleneck size using a 3% variant calling threshold.

this variant is found (**S10C Fig**), the average bottleneck size across all transmission pairs remains low (mean = 9, 95% CI: 6–14).

It is possible that some of the pairs evaluated were not direct transmission pairs. Instead individuals may be part of the same transmission chain or share a common source of infection. We reasoned if two individuals were infected from a common source, then they may have developed symptoms around the same time. In contrast, if one individual infected the other, then their symptom onset dates should be staggered. To assess this, we compared bottleneck sizes to the time between symptom onset in donor-recipient pairs for which symptom onset dates were available (n = 17) (**S11 Fig**). We observed no clear trend between bottleneck size and symptom onset intervals. Finally, all bottleneck estimates are inherently limited by access to a single time point from each donor and recipient. Because it is impossible to know the exact date of infection and transmission, the donor iSNV frequencies may not reflect the true diversity present at the time of transmission. Taken together, we find that even among house-hold pairs, the number of transmitted viruses is small. Although bottleneck size estimates vary by variant calling threshold, we find consistent support for fewer than 50 viruses founding infection, suggesting that the majority of transmission events are founded by very few viruses. Our data suggest that iSNVs generated within-host are generally lost during the transmission event, and are not efficiently propagated among epidemiologically linked individuals.

## Discussion

The emergence of divergent SARS-CoV-2 lineages has called into question the role of within-host selection in propagating novel variants. Our results suggest that very limited variation is generated and transmitted during acute SARS-CoV-2 infection. Most infections in our dataset are characterized by fewer than 5 total intersection iSNVs, the majority of which are low-frequency. Less than half of iSNVs are not detected in global consensus genomes, and are rarely detected in downstream branches on the phylogenetic tree. We show that even among putative household transmission pairs, iSNVs are shared infrequently, and we estimate that a small number of viruses found infection after transmission. The combination of low within-host diversity, tight transmission bottlenecks, and infrequent propagation along transmission chains may slow the rate of novel variant emergence among acutely infected individuals.

Relatively few studies have reported on SARS-CoV-2 within-host diversity, and their results have varied. SARS-CoV-2 within-host sequence data appear to be particularly vulnerable to method error, including sensitivity to cycle threshold [20,21], putative false positive iSNV calls in control runs [20], an uncertain degree of recurrent mutations shared across unrelated samples [21,28,29,39], and variation between technical replicates. Each lab employs its own sample preparation and variant calling pipelines, making comparison across datasets challenging, and concern has been raised regarding recurrent errors that are platform- and lab-specific [40]. Our study is unique in that we designed our protocols with the specific intent of mitigating sources of bioinformatic and laboratory error. To this end, we have attempted to employ multiple, overlapping controls to mitigate errors that could arise from sample preparation, bioinformatic processing, and improper variant thresholds. Our results emphasize the importance

of duplicate sequencing for any studies relying on low-frequency iSNVs to infer biological processes. Like Valesano et al. [20], we observe that SARS-CoV-2 variant calls are sensitive to Ct and variant-calling criteria. We echo their expressed caution in interpreting SARS-CoV-2 within-host data in the absence of pipeline-specific controls.

We find that most samples harbor very few iSNVs and that most variants are low-frequency, in agreement with work from others [20,21,39]. Although we employ distinct methods, we corroborate findings by Lythgoe & Hall et al. [21] that iSNVs do not cluster geographically or phylogenetically, suggesting that they are not transmitted efficiently within communities. We detect a higher number of shared/recurrent iSNVs in our dataset than reported by Lythgoe & Hall et al. [21], Valesano et al. [20], and Shen et al. [39], but fewer than Popa & Genger et al. [28] and James et al. [29]. While some degree of shared iSNVs is reported across most SARS-CoV-2 datasets [20,21,28,29,39] the exact frequency of shared sites is variable, which could be influenced by variant reporting methods and the number of transmission pairs in the dataset. Future work will be necessary to determine the precise degree to which iSNVs recur across unrelated individuals, and the extent to which factors like viral copy number, time of infection, host factors including pre-existing immunity, and sequencing pipeline influence these estimates.

Four other groups have previously estimated the size of the SARS-CoV-2 transmission bottleneck, although the total number of transmission events evaluated to date across studies remains small (~66). Lythgoe & Hall et al. (n = 14 pairs) [41], James & Ngcapu et al. (n = 11 pairs) [29], and Wang et al. (n = 2 pairs) [42] report narrow bottlenecks, in which infection is initiated by fewer than 10 viruses. Popa & Genger et al. (n = 39 pairs) [28] report bottleneck sizes ranging from 10 to 5000, although a reanalysis using a more conservative variant threshold reported a bottleneck estimate of 1–3 virus particles [30]. While current evidence is converging to support a narrow transmission bottleneck for SARS-CoV-2, similar to influenza virus [18,43,44], more data is needed. Our analysis relied on anecdotal evidence of transmission, without epidemiological data to investigate alternative infection histories beyond household membership. It is therefore plausible that some of our transmission pairs were infected independently or from a common source. Future studies examining a larger number of transmission events will be necessary to further refine bottleneck estimates, and to determine whether factors like route of transmission [45] impact transmission bottleneck size.

Our results lead to two main findings. The first is that in individual, acute infections, within-host diversity is limited. The second is that most iSNVs that are present within hosts are lost during transmission. Our results imply that the de novo generation, within-host selection, and subsequent transmission of mutations during the confines of an acute infection is rare. Of course, our dataset represents only a small fraction of the SARS-CoV-2 infections in Wisconsin, and even the combination of every existing within-host dataset represents only a marginal fraction of the SARS-CoV-2 transmission events that have occurred. Indeed, with 17 transmission pairs, our study is currently the second-largest to investigate SARS-CoV-2 evolution within and between hosts. Therefore, while our findings and those from others suggest that evolution is minimal during acute infection, even rare events can occur given sufficient opportunity. No individual study to date, including our own, is sufficiently powered to define how rare these events may be. High rates of SARS-CoV-2 transmission increase the likelihood of rare transmission events in which low-frequency variants are transmitted. Decreasing the number of infections globally through vaccination and non-pharmaceutical interventions are critical for reducing the opportunities for rare evolutionary events to occur. Ongoing assessment of SARS-CoV-2 evolution within and between individuals, especially in the face of growing immunity, will inform our understanding of how novel variants arise and sweep to fixation in the community and beyond.

Our findings are also consistent with, though do not prove, a regime in which most acute infections play a limited role in the generation and spread of new SARS-CoV-2 variants. Prolonged infections permit additional cycles of viral replication, lead to greater accumulation of intrahost mutations [4–8], and allow more time for within-host selection. Even a modest increase in frequency enhances the likelihood of a beneficial variant being transmitted onward, and selection during transmission may further propagate beneficial variants between hosts [25]. It is therefore possible that the rare instances of prolonged infections play an outsize role in novel variant emergence. Future studies that characterize how variants arise, evolve, and transmit during persistent infections will be important for teasing apart the degree with which acute and prolonged infections contribute to global SARS-CoV-2 evolution.

An important limitation of our study is that all of our samples were collected between March and June of 2020. Our analysis therefore represents a snapshot in time prior to the emergence of variants of concern, and before a significant fraction of the population was immune. Dissecting how the fixation of variant of concern mutations and lineages may have impacted the evolutionary patterns of SARS-CoV-2 will be important future work. It is also important to note that patterns of within-host evolution may differ in individuals with vaccine or infection-induced immunity. Whether antigenic escape mutations arise and are selected within individuals with prior immune exposure remains an open question. Understanding the degree to which within-host evolution is shaped by vaccine and infection-induced immunity, both within-host and globally, will be critical for evaluating the pace of SARS-CoV-2 evolution in the future. Finally, while we did not have access to longitudinal samples in this study, characterization of viral populations over time in individual infections will provide superior resolution regarding the dynamicity of iSNVs and their frequencies throughout the course of infection.

Our data, combined with findings from others, suggest that rapid accumulation of novel mutations within-host is not the norm during acute infection. Like influenza viruses, a significant portion of variation generated within one infected host is likely lost during transmission. The combination of within-host limited diversity and tight transmission bottlenecks should slow the pace at which novel, beneficial variants could emerge during transmission among acutely infected individuals. Future studies that compare within-host diversity in individuals with and without SARS-CoV-2 immunity will be necessary to evaluate whether immunity imposes signatures of within-host selection. Finally, given the increasing appreciation for the potential role of long infections to promote variant emergence, within-host data may provide its maximum benefit for dissecting the process of variant evolution during prolonged infections.

## Materials and methods

### Ethics statement

We obtained a waiver of HIPAA Authorization and were approved to obtain all clinical samples used for this study, along with a Limited Data Set by the Western Institutional Review Board (WIRB #1-1290953-1) and the FUE IRB 2016–0605. This limited dataset contains sample collection data and county of collection. Additional sample metadata, e.g. race/ethnicity, were not shared. The limited metadata were classified as non-identifiable, so consent was not obtained due to sample anonymity.

### Sample approvals and sample selection criteria

Samples selected for iSNV characterization were derived from 150 nasopharyngeal (NP) swab samples collected from March 2020 though July 2020, originating from the University of

Wisconsin Hospital and Clinics and the Milwaukee Health Department Laboratories. Submitting institutions provided a cycle threshold (Ct) or relative light unit (RLU) for all samples. Sample metadata, including GISAID and SRA accession identifiers, are available in **S2 Table**.

Diagnostic assays for the samples included in this study were performed at the University of Wisconsin Hospital and Clinical diagnostic laboratory using CDC's diagnostic RT-PCR [46], the Hologic Panther SARS-CoV-2 assay [47], or the Aptima SARS-CoV-2 assay [48].

### Nucleic acid extraction

Viral RNA (vRNA) was extracted from 100 μl of VTM using the Viral Total Nucleic Acid Purification kit (Promega, Madison, WI, USA) on a Maxwell RSC 48 instrument and eluted in 50 μL of nuclease-free H2O.

### Complementary DNA (cDNA) generation and PCR

Complementary DNA (cDNA) was synthesized according to a modified ARTIC Network approach [49,50]. RNA was reverse transcribed with SuperScript IV VILO (Invitrogen, Carlsbad, CA, USA) according to manufacturer guidelines [49,50]. A SARS-CoV-2-specific multiplex PCR for Nanopore sequencing was performed using the ARTIC v3 primers (**S3 Table**). cDNA (2.5 μL) was amplified in two multiplexed PCR reactions using Q5 Hot-Start DNA High-fidelity Polymerase (New England Biolabs, Ipswich, MA, USA).

### TruSeq Illumina library prep and sequencing for minor variants

All Wisconsin surveillance samples were prepped and sequenced by Oxford Nanopore Technologies (details below) and a subset described in this paper were additionally prepped for sequencing on an Illumina MiSeq. These SARS-CoV-2 samples (n = 150) consisted of household pairs as well as a random sampling of the surveillance cohort selective for enhanced iSNV characterization. Amplified cDNA was purified and made compatible for sequencing on an Illumina MiSeq according to the TruSeq Nano DNA manufacturer instructions (Illumina, USA). The average DNA fragment length and purity was determined using the Agilent High Sensitivity DNA kit and the Agilent 2100 Bioanalyzer (Agilent, Santa Clara, CA). Samples were pooled at equimolar concentrations to a final concentration of 4 nM. All libraries were run on a 500-cycle v2 flow cell. The samples included in this study were sequenced across seven distinct MiSeq runs. Each sample was library prepped and sequenced in technical replicate. Replicates were true replicates in that we started from two aliquots taken from the original samples.

### Oxford nanopore library preparation and sequencing for consensus sequences

All consensus-level surveillance sequencing of SARS-CoV-2 was performed using Oxford Nanopore sequencing (n = 3,351) as described previously [51].

### Processing raw ONT data

Sequencing data was processed using the ARTIC bioinformatics pipeline scaled up using on campus computing cores (https://github.com/artic-network/artic-ncov2019). The entire ONT analysis pipeline is available at https://github.com/gagekmoreno/SARS-CoV-2-in-Southern-Wisconsin.

## Processing raw Illumina data

Raw FASTQ files were analyzed using the workflow available in the following GitHub repository– https://github.com/gagekmoreno/SARS_CoV_2_Zequencer. Reads were paired and merged using BBMerge (https://jgi.doe.gov/data-and-tools/bbtools/bb-tools-user-guide/bbmerge-guide/) and mapped to the Wuhan-Hu-1/2019 reference (Genbank accession MN908947.3) using BBMap (https://jgi.doe.gov/data-and-tools/bbtools/bb-tools-user-guide/bbmap-guide/). Mapped reads were imported into Geneious (https://www.geneious.com/) for visual inspection. Variants were called using callvariants.sh (contained within BBMap) and annotated using SnpEff (https://pcingola.github.io/SnpEff/). Variants were called at $\geq$0.01% in high-quality reads (phred score >30) that were $\geq$100 base pairs in length and supported by a minimum of 10 reads. The total minimum read support was set to 10 to generate initial VCF files with complete consensus genomes for the few samples where coverage fell below 100 reads in a few areas. Substantial downstream variant cleaning was performed as outlined below.

## iSNV quality control

BBMap's output VCF files were cleaned using custom Python scripts, which can be found in the GitHub accompanying this manuscript (https://github.com/lmoncla/ncov-WI-within-host). First, any samples without technical replicates were excluded. Next, we discarded all iSNVs which occurred at primer-binding sites (**S3 Table**). These "recoded" VCFs can be found in the GitHub repository in "data/vcfs-recode". We then filtered these recoded VCF files and for variants with (1) 100x coverage; (2) found at $\geq$3% frequency; (3) and found between nucleotides 54 and 29,837 (based on the first and last ARTIC v3 amplicon). We excluded all indels from our analysis, including those that occur in intergenic regions.

We inspected our filtered iSNV datasets across replicate pairs. We visually inspected each replicate pair VCF and plotted replicate frequencies against each other (available in the GitHub repository). This identified a few samples which were outliers for having very limited overlap in their iSNV populations. This could be traced to low coverage or amplicon drop-out in each sample. FASTQs for these samples are available in GenBank, but we have excluded them from downstream analyses presented here (n = 11; tube/filename identifier 65, 124, 125, 303, 316, 1061, 1388, 1103, 1104, 1147, and 1282) (iSNVs in technical replicates are shown for sample 1104 in **S4B Fig**).

We generated one cleaned VCF file by averaging the frequencies found for overlapping iSNVs and discarding all iSNVs which were only found in one replicate. In addition to the SARS-CoV-2 diagnostic swabs, we sequenced a SARS-CoV-2 synthetic RNA control (Twist Bioscience, San Francisco, CA) representing the Wuhan-Hu-1 sequence (Genbank: MN908947.3) in technical replicate at $1 \times 10^6$ template copies per reaction in order to identify spurious variants introducing during library prep and sequencing. We then excluded variants detected in the synthetic RNA control (**S4 Table**) from all downstream analyses. Notably, this filter removed a single variant at nucleotide position 6,669 from our analysis [20]. Finally, within-host variants called at $\geq$50% and <97% frequency comprise consensus-level mutations relative to the Wuhan-Hu-1/2019 reference sequence. To ensure that the corresponding minor variant was reported we report the opposite minor allele at a frequency of 1—the consensus variant frequency. For example, a C to T variant detected at 75% frequency relative to the Wuhan-1 reference was converted to a T to C variant at 25% frequency.

## Processing of the raw sequence data, mapping, and variant calling with the Washington pipeline

To assess the sensitivity of our iSNV calls to bioinformatic pipelines, we generated VCF files using an independent bioinformatic pipeline. Raw reads were assembled against the

SARS-CoV-2 reference genome Wuhan-Hu-1/2019 (Genbank accession MN908947.3; the same reference used for the alternative basecalling method) to generate pileup files using the bioinformatics pipeline available at https://github.com/seattleflu/assembly. Briefly, reads were trimmed with Trimmomatic (http://www.usadellab.org/cms/?page=trimmomatic) [52] in paired end mode, in sliding window of 5 base pairs, discarding all reads that were trimmed to <50 base pairs. Trimmed reads were mapped using Bowtie 2 (http://bowtie-bio.sourceforge.net/bowtie2/index.shtml) [53], and pileups were generated using samtools mpileup (http://www.htslib.org/doc/samtools-mpileup.html). Variants were then called from pileups using varscan mpileup2cns v2.4.4 (http://varscan.sourceforge.net/using-varscan.html#v2.3_mpileup2cns). Variants were called at ≥1% frequency, with a minimum coverage of 100, and were supported by a minimum of 2 reads.

## Phylogenetic analysis

All available full-length sequences from Wisconsin through February 16, 2021 were used for phylogenetic analysis using the tools implemented in Nextstrain custom builds (https://github.com/nextstrain/ncov) [54,55]. Phylogenetic trees were built using the standard Nextstrain tools and scripts [54,55]. We used custom python scripts to filter and clean metadata. A custom "Wisconsin" profile was made to create a Wisconsin-centric subsampled build to include representative sequences. The scripts and output are available at https://github.com/gagekmoreno/Wisconsin-SARS-CoV-2.

## Household pairs permutation test

For household groups, we performed all pairwise comparisons between members of the household, excluding pairs for which the consensus genomes differed by >2 nucleotide changes. We determined this cutoff by modeling the probability that 2 consensus genomes separated by one serial interval differ by n mutations. We model this process as Poisson-distributed with lambda equal to the expected number of substitutions per serial interval, as described previously [36]. We chose to model this expectation using the serial interval rather than the generation interval for the following reason. The sequence data we have represent cases that were sampled via passive surveillance, usually from individuals seeking testing after developing symptoms. Differences in the genome sequences from two individuals therefore represent the evolution that occurred between the sampling times of those two cases. Although neither the serial interval nor the generation interval perfectly matches this sampling process, we reasoned that the serial interval, or the time between the symptom onsets of successive cases, may more accurately capture how the data were sampled. We evaluated probabilities across a range of serial intervals and clock rates. For serial interval, we use the values inferred by He et al, of a mean of 5.8 days with a 95% confidence interval of 4.8–6.8 days [37]. For substitution rate, we employ estimates from Duchene et al, who estimate a mean substitution rate of $1.10 \times 10^{-3}$ substitutions per site per year, with a 95% credible interval of $7.03 \times 10^{-4}$ and $1.15 \times 10^{-3}$ [1]. To model the expectation across this range of values, we evaluate the probabilities for serial intervals at the mean (5.8), as well as for 4, 5, 6, 7, and 8 days, and substitution rates at the mean ($1.10 \times 10^{-3}$) and at the bounds of the 95% credible interval. For each combination of serial interval and substitution rate, we calculate the expected substitutions in one serial interval as: (substitution rate per site per year * genome length/365 days) *serial interval. The results using the mean serial interval (5.8 days) and substitution rate ($1.10 \times 10^{-3}$) are shown in the main text, while the full set of combinations is shown in the supplement. Under this model, the vast majority of consensus genomes derived from cases separated by a single serial interval are expected to differ by ≤2 mutations. The probability that two genomes that are separated by one serial interval

differ by 3 mutations ranges from 0.0016–0.059. Only in the case of an 8 day serial interval with the highest bound of the substitution rate do we infer a probability of 3 mutations that is greater than 0.05. We therefore classified all pairs of individuals from each household that differed by ≤2 consensus mutations and who were tested within 14 days of each other as putative transmission pairs.

To determine whether putative household transmission pairs shared more variants than individuals without an epidemiologic link, we performed a permutation test. At each iteration, we randomly selected a pair of samples (with replacement) and computed the proportion of variants they share as: (2 x total number of shared variants) / (the total number of variants detected among the two samples). For example, if sample A contained 5 iSNVs relative to the reference (Wuhan-1, Genbank accession MN908947.3), sample B harbored 4 iSNVs, and 1 iSNV was shared, then the proportion of sample A and B's variants that are shared would be 2/9 = 0.22. We performed 10,000 iterations in which pairs were sampled randomly to generate a null distribution. We then compared the proportion of variants shared by each putative household transmission pair to this null distribution. The proportion of variants shared by a household pair was determined to be statistically significant if it was greater than 95% of random pairs.

## Transmission bottleneck calculation

The beta-binomial method [38], was used to infer the transmission bottleneck size $N_b$. $N_b$ quantifies the number of virions donated from the index individual to the contact (recipient) individual that successfully establish lineages in the recipient that are present at the sampling time point. The unit of the transmission bottleneck, as described by Sobel-Leonard and Koelle is the number of viruses that found infection in a recipient host following transmission. The beta-binomial method assumes variant sites are independent, which may not be true given that SARS-CoV-2 contains a continuous genome thought to undergo limited recombination [56]. In addition, the beta-binomial method assumes that identical variants found in the index and contact are shared as a result of transmission, though it is possible that identical variants occurring in a donor and a recipient individual occurred independently of one another and are not linked through transmission. We consider this possibility at one site in particular which commonly appears at low frequencies in donor-recipient pairs. Code for estimating transmission bottleneck sizes using the beta-binomial approach has been adapted from the original scripts (https://github.com/koellelab/betabinomial_bottleneck) and is included in the GitHub accompanying this manuscript (https://github.com/lmoncla/ncov-WI-within-host).

We calculated individual transmission bottleneck size estimates for each household transmission pair as were identified in the household permutation test (n = 28). We used the date of symptom onset and/or date of sample collection to assign donor and recipient within each pair. Within each pair, if the date of symptom onset differed by ≥3 days, we assigned the individual with the earlier date as the donor. If this information was unavailable or uninformative (<3 days) for both individuals in a pair, we looked at the date of sample collection and if these dates differed by ≥3 days, we assigned the individual with the earlier date as the donor. If this information was also not available or was not informative (<3 days), we calculated the bottleneck size with each individual as a donor. These bidirectional comparisons are denoted with an "a" or "b" appended to the filename (n = 16 pairs were analyzed bidirectionally). In total, we analyzed 44 pairs (including bidirectional comparisons). Metadata and GISAID accession numbers for each pair are described in **S4 Table.**

Combined transmission bottleneck size estimates (as seen in **Fig 5C**) were estimated as described in the supplemental methods in Martin & Koelle [30]. Briefly, overall transmission

bottleneck sizes were estimated based on the assumption that transmission bottleneck sizes are distributed according to a zero-truncated Poisson-distribution and bidirectional bottleneck estimates were each assigned 50% of the weight in this calculation compared to the unidirectional pairs. Matlab code to replicate the combined bottleneck estimates can be found in the GitHub accompanying this paper (https://github.com/lmoncla/ncov-WI-within-host).

### Enumerating mutations along the phylogeny

We used the global Nextstrain [54] phylogenetic tree (nextstrain.org/ncov/global) accessed on February 24, 2021 to query whether mutations detected within-host are detected on the global tree. We accessed the tree in JSON format and traverse the tree using baltic [57]. To determine the fraction of within-host variants detected on the phylogenetic tree, we traversed the tree from root to tip, gathering each mutation that arose on the tree in the process. For each mutation, we counted the number of times it arose on internal and terminal nodes. We then compared the fraction of times each iSNV identified within-host was detected on an internal node vs. a terminal node. To determine whether particular iSNVs were enriched at internal nodes, we compared the frequency of that iSNV's detection against the overall ratio of mutations arising on internal vs. terminal nodes in the phylogeny with a Fisher's exact test.

To query whether iSNVs ever became dominant in tips sampled downstream, we used a transmission metric developed previously [58]. Using the tree JSON output from the Nextstrain pipeline [54], we traversed the tree from root to tip. We collapsed very small branches (those with branch lengths less than $1 \times 10^{-16}$) to obtain polytomies. For each tip for which we had within-host data that lay on an internal node, i.e., had a branch length of nearly 0 ($< 1 \times 10^{-16}$), we then determined whether any subsequent tips occurred in the downstream portion of the tree, i.e., tips that fall along the same lineage but to the right of the parent tip. We then traversed the tree and enumerated every mutation that arose from the parent tip to each downstream tip. If any mutations along the path from the parent to downstream tip matched a mutation found within-host in the parent, this was classified as a potential instance of variant tfransmission. A diagram of how "downstream tips" and mutations were classified is shown in **Fig 4A**.

### Linear regression model

To determine the relative contributions of phylogenetic divergence, geographic distance, clade membership, and household membership to the probability of sharing within-host variants, we fit linear regression models to the data in R. As our outcome variable, we performed pairwise comparisons for each pair of samples in the dataset (including household and non-household pairs) and compute the proportion of variants shared for each pair. We then model the proportion of shared variants as the combined function of 4 predictor variables as follows: Proportion of variants shared $\sim \beta_0 + \beta_1 x_1 + \beta_2 x_2 + \beta_3 x_3 + \beta_4 x_4$, where $x_1$ represents a 0 or 1 value for household, where a 1 indicates the same household and a 0 indicates no household relationship. $X_2$ denotes the divergence, i.e., the branch length in mutations between tip A and tip B as a continuous variable, $x_3$ indicates the great circle distance in kilometers between the location of sample collection as a continuous variable, and $x_4$ denotes a 0 or 1 for whether the two tips belong to the same clade (same clade coded as a 1, different clade coded as a 0). We fit a univariate model for each variable independently, a model with an intercept alone, and a combined model using the Rethinking package in R (https://www.rdocumentation.org/packages/rethinking/versions/1.59). We perform model comparison with the WAIC metric and select the combined model as the one with the best fit. We compute mean coefficient estimates and 95% highest posterior density intervals (HPDI) by sampling and summarizing 10,000 values from the posterior distribution.

## Supporting information

**S1 Fig. Read depth.** Read depth by genome location in 1,000-bp bins for MiSeq runs **a.** 627, **b.** 628, **c.** 643, and **d.** 644. Water controls and low-coverage samples are labeled. Samples included in each run are labeled according to the color to the right of each plot.
(PDF)

**S2 Fig. Read depth.** Read depth by genome location in 1,000-bp bins for MiSeq runs **a.** 645, **b.** 667, and **c.** 671. Water controls and low-coverage samples are labeled within each plot. Samples included in each run are labeled according to the color to the right of each plot.
(PDF)

**S3 Fig. Additional iSNV quality control information.** Subplot **a.** shows variant frequencies generated using the Wisconsin bioinformatic pipelines are shown on the x-axis and frequencies generated using the Washington bioinformatic pipeline are shown on the y-axis. The yellow box highlights low-frequency variants (3–15%), which is expanded out to the right. **b.** Proportion of intersection iSNVs relative to the total number of iSNVs increases as variant frequency threshold increases. **c.** The total number of iSNVs detected across both Twist RNA control replicates compared to the iSNV frequency threshold. 57/102 of iSNVs detected in these clonal samples occur <3% frequency. Note that the iSNVs reported in **S1 Table** are intersection iSNVs only. The identities of all iSNVs detected ≥1% frequency in the Twist RNA control can be found in the GitHub accompanything this manuscript. **d.** The number of intersection variants, both consensus and iSNVs, is compared to the Ct value for all samples where a Ct value was available. Out of 133 total samples, Ct values were available for 94. **e.** The number of intersection variants, both consensus and iSNVs, is compared to the RLU (relative light unit) value for all samples where a RLU value was available.
(PDF)

**S4 Fig. iSNVs in technical replicates across all samples. a.** Variant frequencies in replicate 1 are shown on the x-axis and frequencies in replicate 2 are shown on y-axis. This plot includes all variants found in both replicates and not just the intersection variants as shown **Fig 1A and 1B.** Example of one sample with very poor overlap between technical replicates; this sample (sample 1104) was excluded from the experimental dataset.
(PDF)

**S5 Fig. iSNVs do not cluster by sequencing run.** iSNVs detected in at least 2 samples are shown on the x-axis and are plotted against the number of times they are detected in our dataset. Each iSNV bar is colored according to the number of times it was detected within each sequencing batch.
(PDF)

**S6 Fig. Wisconsin divergence phylogeny.** A full-genome phylogenetic tree built showing 6306 Wisconsin consensus sequences with the Nextstrain pipeline is shown. The x-axis represents divergence expressed as the number of nucleotide mutations. Nextstrain clade labels are shown on the corresponding branch. Yellow tips represent Wisconsin samples that were Illumina sequenced in duplicate and analyzed in this manuscript. Purple tips represent samples from households.
(PDF)

**S7 Fig. Most iSNVs are not detected on the phylogeny.** We queried every iSNV that was detected within-host (in at least 1 sample) in the global SARS-CoV-2 phylogenetic tree and quantified the number of times that iSNV was detected on an internal node (yellow bar

heights) or on a terminal node/tip (blue bar heights). 42% of all iSNVs detected within-host were found on the tree. Most iSNVs that were detected on the tree were rare, and occurred predominantly on terminal nodes. Please note you will likely need to zoom into this figure to clearly read the labels along the x-axis.
(PDF)

**S8 Fig. Modeling the expected number of mutations distinguishing genomes separated by one serial interval.** To define whether infections sampled from the same household might be true transmission pairs, we explored the expected number of consensus mutations that should differ between genomes separated by one serial interval. We modeled the probability that 2 consensus genomes will share x mutations as Poisson distributed with lambda equal to the number of mutations expected to accumulate in the SARS-CoV-2 genome over a single serial interval, given a known substitution rate. He et al. estimate a serial interval for SARS-CoV-2 of of 5.8 days, with a 95% confidence interval between 4.8–6.8 days [35]. We therefore evaluated serial intervals of 4, 5, 6, 7, and 8 days. For the substitution rate, we use estimates from Duchene et al [1], who estimate a mean substitution rate of $1.10 \times 10^{-3}$ substitutions per site per year, with a 95% credible interval of $7.03 \times 10^{-4}$ and $1.15 \times 10^{-3}$. We evaluated the probabilities that two consensus genomes differ by 0, 1, 2, 3, and 4 mutations given serial intervals ranging from 4–8, and clock rates at the mean, and upper and lower bounds of the 95% credible interval. For each calculated probability, the serial interval is represented by color and the substitution rate is shown above each plot. The dotted line represents a probability of 0.05. Given these combinations of values, the vast majority of consensus genomes are expected to differ by 0–2 mutations.
(PDF)

**S9 Fig. Posterior density estimates for regression coefficients.** For each regression coefficient evaluated in the combined regression model, the full posterior distribution is shown as a density plot. The posterior distribution of the estimated variance and intercept are also shown.
(PDF)

**S10 Fig. Sensitivity testing of transmission bottleneck estimates.** Maximum likelihood estimates for mean transmission bottleneck size in individual donor-recipient pairs using **a.** 1% frequency threshold, **b.** 3% frequency threshold, **c.** excluding site 15,168 as a possible homoplasy with a 3% frequency threshold, and **d.** 7% frequency threshold. Data are not shown for donor-recipient pairs where no bottleneck estimate could be generated due to lack of variant data. Bidirectional comparisons are indicated with an "a" and "b" following the pair number.
(PDF)

**S11 Fig. Variance in transmission bottleneck size cannot be explained by time between symptom onset in donor:recipient pairs.** We plotted transmission bottleneck size on the y-axis against time (days) between symptom onset in 17 donor-recipient pairs on the x-axis for which we had symptom metadata.
(PDF)

**S12 Fig. Frequency of SNVs across the genome.** The frequency of all SNVs are plotted across the SARS-CoV-2 genome. Each variant is colored by mutation type.
(PDF)

**S1 Table. iSNVs detected in replicate sequencing of the synthetic RNA control (Twist-Biosciences).** All iSNVs called in the synthetic RNA control from Twist Biosciences are shown.
(DOCX)

**S2 Table. Sample identifiers and accession numbers.** This table includes strain name, tube/filename, state of collection, county of collection, collection date, GISAID accession number, Genbank accession number, as well as Ct values and RLU values where available for each sample included in this study.
(DOCX)

**S3 Table. ARTIC v3 primer sequences used to amplify cDNA for library preparation.**
(DOCX)

**S4 Table. Household transmission pair metadata including accession numbers, difference in days between symptom onset, difference in days between collection dates, and pair identifier.**
(DOCX)

**S5 Table. List of all iSNVs and their respective frequencies.**
(PDF)

## Author Contributions

**Conceptualization:** Katarina M. Braun, Gage K. Moreno, Cassia Wagner, Katia Koelle, David H. O'Connor, Thomas C. Friedrich, Louise H. Moncla.

**Data curation:** Katarina M. Braun, Molly A. Accola, William M. Rehrauer, David A. Baker, David H. O'Connor, Louise H. Moncla.

**Formal analysis:** Katarina M. Braun, Gage K. Moreno, David A. Baker, Katia Koelle, Louise H. Moncla.

**Funding acquisition:** David H. O'Connor, Thomas C. Friedrich.

**Investigation:** Katarina M. Braun, Gage K. Moreno, Louise H. Moncla.

**Methodology:** Gage K. Moreno, Molly A. Accola, Katia Koelle, Trevor Bedford, Thomas C. Friedrich, Louise H. Moncla.

**Resources:** Katia Koelle, Thomas C. Friedrich.

**Software:** Katarina M. Braun, Gage K. Moreno, Cassia Wagner, David A. Baker, Katia Koelle, Louise H. Moncla.

**Supervision:** David H. O'Connor, Trevor Bedford, Thomas C. Friedrich, Louise H. Moncla.

**Validation:** Katarina M. Braun, Gage K. Moreno, Cassia Wagner, Louise H. Moncla.

**Visualization:** Katarina M. Braun, Gage K. Moreno, Louise H. Moncla.

**Writing – original draft:** Katarina M. Braun, Gage K. Moreno, Cassia Wagner, Katia Koelle, Thomas C. Friedrich, Louise H. Moncla.

**Writing – review & editing:** Katarina M. Braun, Gage K. Moreno, Thomas C. Friedrich, Louise H. Moncla.

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
