## [Decision Letter · Decision Letter 0]

25 Jun 2021

Dear Ms. Braun,

Thank you very much for submitting your manuscript "Limited within-host diversity and tight transmission bottlenecks limit SARS-CoV-2 evolution in acutely infected individuals" for consideration at PLOS Pathogens. As with all papers reviewed by the journal, your manuscript was reviewed by members of the editorial board and by several independent reviewers. In light of the reviews (below this email), we would like to invite the resubmission of a significantly-revised version that takes into account the reviewers' comments.

You will see that both reviewers are generally satisfied with the technical aspects of the work but have concerns about the interpretation of the results and the overstatement of the conclusions. These should be addressed in a revision with appropriate qualifications placed on the conclusions.

We cannot make any decision about publication until we have seen the revised manuscript and your response to the reviewers' comments. Your revised manuscript is also likely to be sent to reviewers for further evaluation.

Sincerely,

Scott C. Weaver

Guest Editor

PLOS Pathogens

David Wang

Section Editor

PLOS Pathogens

Kasturi Haldar

Editor-in-Chief

PLOS Pathogens

orcid.org/0000-0001-5065-158X

Michael Malim

Editor-in-Chief

PLOS Pathogens

orcid.org/0000-0002-7699-2064

You will see that both reviewers are generally satisfied with the technical aspects of the work but have concerns about the interpretation of the results and the overstatement of the conclusions. These should be addressed in a revision with appropriate qualifications placed on the conclusions.

Reviewer's Responses to Questions

**Part I - Summary**

Reviewer #1: This study looks to understand the intra-host diversity (iSNVs) and transmission bottlenecks of SARS-CoV-2 during acute infection. The authors sequence SARS-CoV-2 genomes present in nasal swabs from acutely infected individuals obtained March to July 2020. They apply an array of well thought out controls to analyze their data and conclude that there is little diversity within hosts and the iSNVs present do not contribute significantly to global consensus changes. Moreover, they analyze household transmission pairs and conclude that transmission bottlenecks are likely narrow. Together, the authors conclude that due to low intra-host diversity and narrow transmission bottlenecks, SARS-CoV-2 evolution or emergence is limited in acutely infected individuals.

Reviewer #2: This is an interesting study, which address an important question. In general, the study is technically sounded, but the authors should be more careful at the interpretations of the results and general implications. It would make a stronger report if the authors explicitly recognize the limitation of the study (i.e. number of patients, depth of the sequencing data, etc). See specific comments, but I also recommend to carefully revise the text to provide a more complete report of the caveat on the current study.

**Part II – Major Issues: Key Experiments Required for Acceptance**

Reviewer #1: This study is technically well done with extreme care in their data analysis and controls. I think this work is important and confirms previously published studies. However, I’m finding it hard to see the absolute novelty of the conclusions when comparing this study to several others the authors mention. In addition, there are several major points to be addressed for publication.

1. My major concern with this study are with the conclusions drawn from the data. First, the title of this manuscript implies mechanism. As written, it states that limited within-host diversity actually limits SARS-CoV-2 evolution. I don’t think the authors have shown that here and from the manuscript they think the limited diversity implies the limited evolution. The title should be changed to show what the study found. Maybe something like “Individuals acutely infected with SARS-CoV-2 have limited within-host diversity and tight transmission bottlenecks”.

Second, the way the manuscript is written, the authors conclusions are not very defined. For example, they write “Variation is shared among some household samples, but is likely insufficient…” “transmission bottlenecks are likely narrow”. Writing it this way makes me wonder, is it or isn’t it? There are a few other vague statements throughout the manuscript “Figure 4: Household pairs share a modest degrees of within-host variation”. What does modest degree mean? The authors did a great job analyzing the data and it’s correct, the conclusions should be strong.

Last, some conclusions don’t seem to match up and the authors don’t justify why they conclude these things. For example, in the section “Most within-host variation does not contribute to consensus diversity” the authors state that 42% of iSNVs they find are found globally, yet in the caption of Supplemental Figure 7 they state that 1/3rd of iSNVs were detected. Which is it? I suppose that because this number is less than 50% the conclusion is correct but to me 42% is a lot and suggests that these iSNVs are out there.

In addition, they conclude in the discussion “our results imply that the accumulation of multiple iSNVs is unlikely during typical, acute infection” Why do you conclude that? In figure 1 you show that many individuals have 2+ mutations. Is there a reason for this conclusion beyond the data? I could look at these data as even though most iSNVs are not found globally and most people have 1 iSNV, there are still many people with multiple iSNVs and lots of iSNVs found globally.

I suggest editing the manuscript significantly to make sure all your conclusions are backed up by your strong data. On this note, try to distinguish your study from others done before. I found this hard to see reading through.

2. I found the way the iSNVs were presented a bit confusing. First, are there consensus changes in individuals? From the data, yes there are. The authors call them “Wuhan reversions” but they are actually consensus changes. These changes are very important but they are not talked about in Figure 1d. Along these same lines, in Figure 2, it would be very helpful to know the frequency of the variants in 2b. I think a Table showing all of the variants found, their frequencies, and their protein changes would be very helpful. It would also be helpful to have a genome schematic in the figure(s) to show where exactly on the genome each SNP is found.

3. Finally, one thing that is important to address is how specific do you think these conclusions are to the samples you analyzed? You are analyzing swabs from March-July 2020, right at the beginning of the pandemic. If I remember, the D614G mutation is not in your reference genome and you don’t find it yet. So SARS-CoV-2 has not really started evolving. Do you think if you did the same analysis with samples from 2021 you’d get the same results? Maybe another study did this? I think this is important to address, as it is possible that an emerging variant in the SARS-CoV-2 nonstructural proteins could have changed the fidelity of SARS-CoV-2 to increase evolution. This is merely hand waving but I’m curious as to whether SARS-CoV-2 has increased its evolutionary capacity by adaptive mutations not in the Spike (which we tend to focus on). So maybe at the beginning of the pandemic your conclusions are correct yet, as the pandemic evolved, so did SARS-CoV-2 diversity/transmission bottlenecks. How your conclusions hold up over time would be important to discuss.

Reviewer #2: The authors should remove the phrase “enhanced transmissibility” because this has not been thoroughly demonstrated for any of the variants. The main observations about variants of concern are their ability to replace some of the older variants in the population and/or evade to some degree neutralization by circulating antibodies. The 2 references (2 and 3) in the paper actually say the same thing.

Could the authors describe the dataset from households? In particular, how many samples/household? Were there more than one sample/individual? At what day post symptom onset? Also, in order to understand the rate of variant emergence during acute infection, the authors should detail when samples were collected after symptom onset.

What is the unit of bottleneck size?

Overall, the analysis done in this study is well controlled but one main question regarding the sample size of the study remains: Is the dataset big enough to rule out intrahost diversity’s role in variant emergence? A few chronic infections might play a major role in variant emergence but considering the large numbers of acute infections in the world population, even rare events in acute infection might play a significant role in it too.

The study is technically well done and the controls to identify iSNV are robust. I think the discussion is lacking focus on some of the study limitation like not knowing the viral input, the small sample size and not having epidemiological evidence on transmission pairs.

For instance, are infections with very high viral loads likely to harbor more diversity. The authors suggest that this is not the case, but this seems to be counterintuitive.

As the samples were from public health surveillance, they are taken at a single time point in the infection. We still don’t know with this dataset if diversity changes over time in an individual and if this will impact on the capacity for non-dominant variants to be transmitted.

The limitations of the transmission pair permutation test are not discussed. How certain are we of the directionality of transmission correct?

In general the text is long and could be made more brief and to the point.

**Part III – Minor Issues: Editorial and Data Presentation Modifications**

Reviewer #1: (No Response)

Reviewer #2: see previous section

PLOS authors have the option to publish the peer review history of their article (what does this mean?). If published, this will include your full peer review and any attached files.

Reviewer #1: No

Reviewer #2: No
---

## [Editor Report · Decision Letter 1]

29 Jul 2021

Dear Dr. Moncla,

We are pleased to inform you that your manuscript 'Acute SARS-CoV-2 infections harbor limited within-host diversity and transmit via tight transmission bottlenecks' has been provisionally accepted for publication in PLOS Pathogens.

Best regards,

Scott C. Weaver

Guest Editor

PLOS Pathogens

David Wang

Section Editor

PLOS Pathogens

Kasturi Haldar

Editor-in-Chief

PLOS Pathogens

orcid.org/0000-0001-5065-158X

Michael Malim

Editor-in-Chief

PLOS Pathogens

orcid.org/0000-0002-7699-2064
---

## [Editor Report · Acceptance letter]

11 Aug 2021

Dear Dr. Moncla,

We are delighted to inform you that your manuscript, "Acute SARS-CoV-2 infections harbor limited within-host diversity and transmit via tight transmission bottlenecks," has been formally accepted for publication in PLOS Pathogens.

Best regards,

Kasturi Haldar

Editor-in-Chief

PLOS Pathogens

orcid.org/0000-0001-5065-158X

Michael Malim

Editor-in-Chief

PLOS Pathogens

orcid.org/0000-0002-7699-2064